# FIELD-DIT: DIFFUSION TRANSFORMER ON UNIFIED VIDEO, 3D, AND GAME FIELD GENERATION

**Kangfu Mei**
Johns Hopkins University
kmei1@jhu.edu

**Mo Zhou**
Johns Hopkins University
mzhou32@jhu.edu

**Vishal M. Patel**
Johns Hopkins University
vpatel36@jhu.edu

## ABSTRACT

The probabilistic field models the distribution of continuous functions defined over metric spaces. While these models hold great potential for unifying data generation across various modalities, including images, videos, and 3D geometry, they still struggle with long-context generation beyond simple examples. This limitation can be attributed to their MLP architecture, which lacks sufficient inductive bias to capture global structures through uniform sampling. To address this, we propose a new and simple model that incorporates a view-wise sampling algorithm to focus on local structure learning, along with autoregressive generation to preserve global geometry. It adapts cross-modality conditions, such as text prompts for text-to-video generation, camera poses for 3D view generation, and control actions for game generation. Experimental results across various modalities demonstrate the effectiveness of our model, with its 675M parameter size, and highlight its potential as a foundational framework for scalable, architecture-unified visual content generation for different modalities with different weights. Our project page can be found at https://kfmei.com/Field-DiT/.

## 1 INTRODUCTION

Generative tasks (Rombach et al., 2022; Ramesh et al., 2022) are overwhelmed by diffusion probabilistic models that hold state-of-the-art results on most modalities like audio, images, videos, and 3D geometry. Take image generation as an example, a typical diffusion model (Ho et al., 2020) consists of a forward process for sequentially corrupting an image into standard noise, a backward process for sequentially denoising a noisy image into a clear image, and a score network that learns to denoise the noisy image.

The forward and backward processes are agnostic to different data modalities; however, the architectures of the existing score networks are not. The existing score networks are highly customized towards a single type of modality, which is challenging to adapt to a different modality. For example, a recently proposed multi-frame video generation network (Ho et al., 2022b;a) adapting single-frame image generation networks involves significant designs and efforts in modifying the score networks.

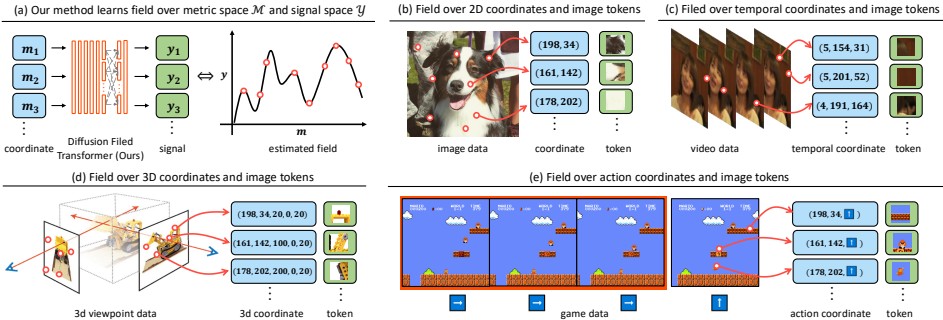

Figure 1: Illustration of the field model's capability to model visual content. The model learns the distribution through attention between coordinate-signal pairs, which is modality-agnostic.

Therefore, it is important to develop a single, versatile architecture that can be applied across different modalities without modification. Such a unified architecture simplifies the development process, reduces the complexity associated with designing modality-specific models, and enables knowledge transfer between modalities.

Field model (Sitzmann et al., 2020; Tancik et al., 2020; Dupont et al., 2022b; Zhuang et al., 2023) is a promising unified score network architecture for different modalities. It learns the distribution over the functional view of data. Specifically, the field $f$ maps the observation from the *metric* space $\mathcal{M}$ (*e.g.*, coordinate or camera pose) into the *signal* space $\mathcal{Y}$ (*e.g.*, RGB pixel) as $f : \mathcal{M} \mapsto \mathcal{Y}$. For instance, an image is represented as $f : \mathbb{R}^2 \mapsto \mathbb{R}^3$ that maps the spatial coordinates (*i.e.*, height and width) into RGB values at the corresponding location, while a video is represented as $f : \mathbb{R}^3 \mapsto \mathbb{R}^3$ that maps the spatial and temporal coordinates (*i.e.*, frame, height, and width) into RGB values. Different modalities usually use different weights to ensure the best performance. Recently, diffusion models are leveraged to characterize the field distributions over the functional view of data (Zhuang et al., 2023) for field generation. Given a set of coordinate-signal pairs $\{(\boldsymbol{m}_i, \boldsymbol{y}_i)\}$, the field $f$ is regarded as the score network for the backward process, which turns a noisy signal into a clear signal $\boldsymbol{y}_i$ in a sequential process with $\boldsymbol{m}_i$ being fixed all the time. The visual content is then composed of the clear signal generated on a grid in the metric space.

Nevertheless, diffusion-based field models for generation still lag behind the modality-specific approaches (Dhariwal & Nichol, 2021; Ho et al., 2022b; He et al., 2022) for learning from dynamic data in high resolution (Bain et al., 2021; Yu et al., 2023a). For example, a 240p video lasting 5 seconds is comprised of up to 10 million coordinate-signal pairs. Due to the memory bottleneck in existing GPU-accelerated computing systems, recent field models (Zhuang et al., 2023) are limited to observe merely a small portion of these pairs (*e.g.*, $1\%$) that are uniformly sampled during training. This limitation significantly hampers the field models in approximating distributions from such sparse observations (Quinonero-Candela & Rasmussen, 2005). Consequently, diffusion-based field models often struggle to capture the fine-grained local structure of the data, leading to, *e.g.*, unsatisfactory blurry results.

While it is possible to change the pair sampling algorithm to sample densely from local areas instead of uniformly, the global geometry is weakened. To alleviate this issue, it is desirable to introduce some complementary guidance on the global geometry in addition to local sampling.

In this paper, we propose a new diffusion field transformers, called **Field-DiT**. In contrast to previous methods, Field-DiT preserves both the local structure and the global geometry of the fields during learning by employing a new view-wise sampling algorithm in the coordinate space, and incorporates additional inductive biases from the text descriptions and autoregressive generation. By combining these advancements with our simplified transformer architecture, we demonstrate that modeling capability of our model surpasses previous methods, achieving improved generated results under the same memory constraints. We empirically validate its superiority against previous domain-agnostic methods across three different tasks, including text-to-video generation, 3D novel-view generation, and game generation. Various experiments show that our method achieves compelling performance even when compared to the state-of-the-art domain-specific methods, underlining its potential as a scalable and architecture-unified visual content generation model across various modalities. Our contributions are summarized as follows:

- We propose a new transformer-based diffusion field model for long-context modeling, which comprises of a view-wise sampling algorithm and autoregressive generation for local structure and global geometry model respectively.
- We demonstrate the effectiveness and efficiency of a simple 675M model on different modalities generation including video, 3D, and game in a unified-architecture, which largely closes the performance gap with modality-specific models with different weights.
- We show the potential of action game generation using diffusion models, and we release the benchmarks including both training and testing data for replication and comparisons.

## 2 RELATED WORK

**Generation Models.** In recent years, generative models have shown impressive performance in visual content generation. The major families are generative adversarial networks (Goodfellow et al.,

2020; Mao et al., 2017; Karras et al., 2019; Brock et al., 2019), variational autoencoders (Kingma & Welling, 2014; Vahdat & Kautz, 2020), auto-aggressive networks (Chen et al., 2020; Esser et al., 2021), diffusion models (Ho et al., 2020; Song et al., 2021), and consistency models (Mei et al., 2024a; Song et al., 2023). Recent diffusion models have obtained significant advancement with stronger network architectures (Dhariwal & Nichol, 2021; Mei et al., 2025b), additional text and image condition (Ramesh et al., 2022; Mei et al., 2024b; 2025a), and pretrained latent space (He et al., 2022). Our method built upon these successes and targets at scaling domain-agnostic models.

**Field Models.** Field models like SIREN Sitzmann et al. (2020) excel at effectively handling diverse data types, such as images, videos, 3D shapes, and audio, without requiring extensive customization. Compared with the modality-specific models, field models enable scalability by allowing advancements in one domain (e.g., images) to directly enhance others (e.g., 3D modeling and video synthesis), streamlining research and development. In order to model complex field distributions, representative methods like Functa (Dupont et al., 2022b) and GEM (Du et al., 2021) adopt a two-stage modeling paradigm: first parameterizing fields, then learning distributions over the parameterized latent space. However, the learning efficiency of the two-stage methods hinders scaling the models, as their first stage incurs substantial computational costs to compress fields into latent codes. Building on recent exploration Zhuang et al. (2023) into the use of diffusion models, which are more powerful for directly modeling complex data distributions without additional parametrization, we propose to model field distributions using explicit coordinate-signal pairs. Nevertheless, field models struggle with very large or highly diverse datasets, such as high-resolution videos. This is due to the complexity of preserving both local structures and global geometry. In contrast, our method leverages the benefits of a single-stage modeling approach, improving accuracy in preserving both local structures and global geometry.

**Long-context Modeling.** Our method also differs from the recently proposed domain-specific works for high-resolution, dynamic data, which models specific modalities in a dedicated latent space, including Spatial Functa (Bauer et al., 2023) and PVDM (Yu et al., 2023c). These methods typically compress the high-dimensional data into a low-dimensional latent space. However, the compression is usually specific to a center modality and lacks the flexibility in dealing with different modalities. For instances, PVDM compresses videos into three latent codes that represent spatial and temporal dimensions separately. However, such a compressor cannot be adopted into the other similar modalities like 3D scenes. In contrast, our method owns the unification flexibility and the achieved advancement can be easily transferred into different modalities.

## 3 METHOD

**Definition.** Conceptually, the diffusion-based field models sample from field distributions by reversing a gradual noising process. As shown in Fig. 1, in contrast to the data formulation of the conventional diffusion models (Ho et al., 2020) applied to the complete data like a whole image, diffusion-based field models apply the noising process to the sparse observation of the field, which is a kind of parametrized functional representation of data consisting of coordinate-signal pairs, *i.e.*, $f : \mathcal{M} \mapsto \mathcal{Y}$. Specifically, the sampling process begins with a coordinate-signal pair $(\mathbf{m}_i, \mathbf{y}_{(i,T)})$, where the coordinate comes from a field and the signal is a standard noise, and less-noisy signals $\mathbf{y}_{(i,T-1)}, \mathbf{y}_{(i,T-2)}, \ldots$, are progressively generated until reaching the final clear signal $\mathbf{y}_{(i,0)}$, with $\mathbf{m}_i$ being constant. Diffusion Probabilistic Field (DPF) (Zhuang et al., 2023) is one of the recent representative diffusion-based field models. It parameterizes the denoising process with a transformer-based network $\epsilon_\theta(\cdot)$, which takes noisy coordinate-signal pairs as input and predicts the noise component $\epsilon$ of $\mathbf{y}_{(i,t)}$. The less-noisy signal $\mathbf{y}_{(i,t-1)}$ is then sampled from the noise component $\epsilon$ using a denoising process (Ho et al., 2020).

In practice, when handling low-resolution data consisting of $N$ coordinate-signal pairs with DPF, the scoring network $\epsilon_\theta(\cdot)$ takes all pairs $\{(\mathbf{m}_i, \mathbf{y}_{(i,T)})\}$ as input at once. For high-resolution data with a large number of coordinate-signal pairs that greatly exceed the modern GPU capacity, (Zhuang et al., 2023) uniformly sample a subset of pairs from the data as input. They subsequently condition the diffusion model on the other non-overlapping subset, referred to as *context pairs*. Specifically, the sampled pairs interact with the query pairs through cross-attention blocks. (Zhuang et al., 2023) show that the ratio between the context pairs and the sampling pairs is strongly related to the quality

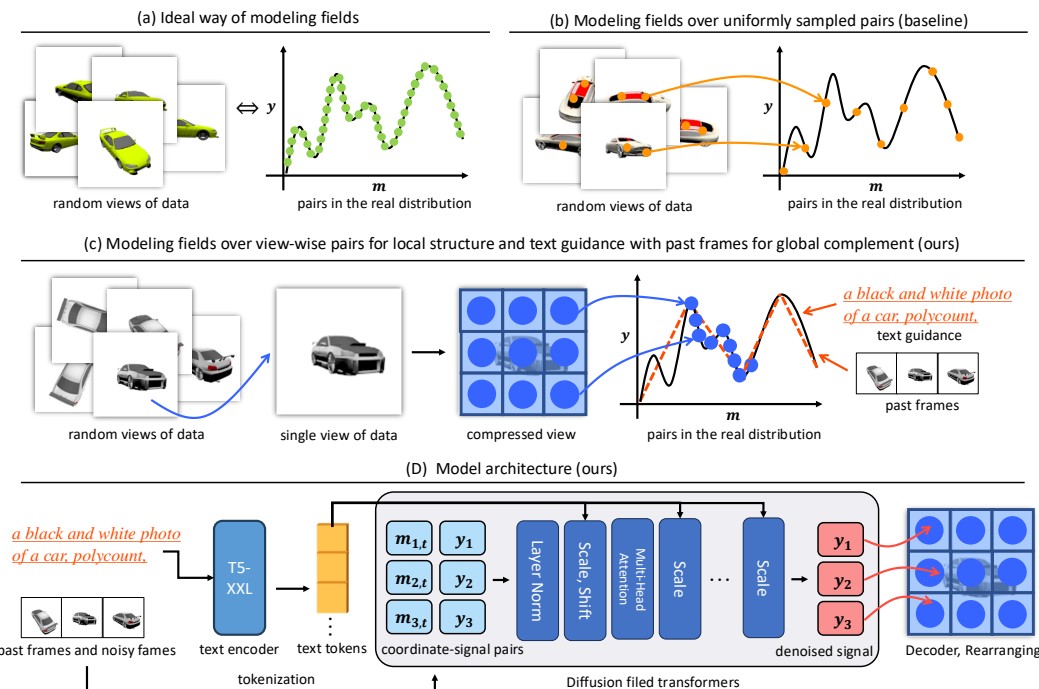

Figure 2: (a) Ideally, all pairs within a field (green points) should be used for training, but this is impractical due to memory limitations. (b) Previous methods uniformly sample a sparse set of pairs (orange points) to represent the field to mitigate memory limitations. (c) Compared to uniform sampling, our local sampling extracts high-fidelity pairs (blue points), better covering the local structure. The text prompt and past frames serve as an approximation to complement the global geometry. (d) Visualization of our sampling pipeline. Note that the input coordinates include the diffusion timesteps of each input frames.

of the generated fields, and the quality decreases as the context pair ratio decreases. Due to the practical memory bottleneck, DPF can only support a maximum $64 \times 64$ resolution, let alone being extended to long context such as multi-frame video generation.

### 3.1 DIFFUSION FIELD TRANSFORMER

In order to scale diffusion-based field models for high-resolution, dynamic data generation, we build upon the recent DPF model (Zhuang et al., 2023) and address its limitations in preserving the local structure of fields, as it can hardly be captured when the uniformly sampled coordinate-signal pairs are too sparse. Specially, our method not only can preserve the local structure, but also introduce additional inductive biases for capturing the global geometry, such as text descriptions and past frames in autoregressive generation.

In order to preserve the local structure of fields, we propose a new view-wise sampling algorithm that samples local coordinate-signal pairs for better representing the local structure of fields. For instance, the algorithm samples the coordinate-signal pairs belonging to a single or several ($n \geqslant 1$; $n$ denotes the number of views) views for video data, where a view corresponds to a single frame. It samples pairs belonging to a single or several rendered images for 3D viewpoints, where a view corresponds to an image rendered at a specific camera pose. A view of an image is the image itself.

This approach restricts the number of interactions among pairs to be modeled and reduces the learning difficulty on high-resolution, dynamic data. Nevertheless, even a single high-resolution view , *e.g.*, in merely $128 \times 128$ resolution) can still consist of 10K pairs, which in practice will very easily reach the memory bottleneck if we leverage a large portion of them at one time, and hence hinder scaling the model for generating high-resolution dynamic data.

To address this issue, our method begins with increasing the signal resolution of coordinate-signal pairs and hence reducing memory usage in the score network. Specifically, we replace the signal

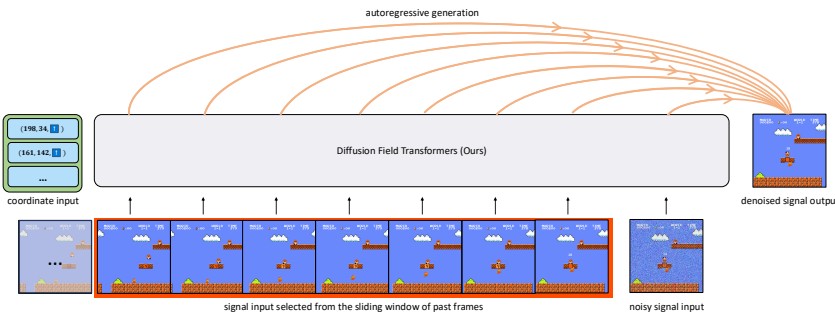

Figure 3: Autoregressive next-frame prediction. Our model takes past frames selected from a sliding window and next action coordinates, such as actions like jump or move, as input. It then generates the next frame, reflecting both the action and the long context of the past frames.

space with a compressed latent space, and employ a more efficient network architecture that only contains decoders. This improvement in efficiency allows the modeling of interactions among pairs representing higher-resolution data while keeping the memory usage constrained. Based on this, one can then model the interactions of pairs within a single or several views of high-resolution data. The overall diagram of the proposed sampling method can be found in Fig. 2.

**View-wise Sampling.** Based on the high-resolution signal and decoder-only network architecture, our method represents field distributions by using view-consistent coordinate-signal pairs, *i.e.*, collections of pairs that belong to a single or several ($n \geq 1$) views of the data, such as one or several frames in a video, and one or several viewpoints of a 3D geometry. In particular, take the spatial and temporal coordinates of a video in $H \times W$ resolution lasting for $T$ frames as an example, for all coordinates $\{\mathbf{m}_1, \mathbf{m}_2, \ldots, \mathbf{m}_i, \ldots, \mathbf{m}_{H \times W \times T}\}$, we randomly sample a consecutive sequence of length $H \times W$ that correspond to a single frame, *i.e.*, $\{\mathbf{m}_1, \mathbf{m}_2, \ldots, \mathbf{m}_i, \ldots, \mathbf{m}_{H \times W}\}$. For data consisting of a large amount of views (*e.g.* $T >> 16$), we randomly sample $n$ views (sequences of length $H \times W$), resulting in an $H \times W \times n$ sequence set. Accordingly, different from the transformers in previous works (Zhuang et al., 2023) that model interaction among all pairs across all views, ours only models the interaction among pairs that belongs to the same view, which reduces the complexity of field model by limiting the number of interactions to be learned.

## 3.2 LONG-CONTEXT CONDITIONING

To complement our effort in preserving local structures that may weaken global geometry learning, since the network only models the interaction of coordinate-signal pairs in the same view, we propose to supplement the learning with a long-context conditioning of the field, avoiding issues in cross-view consistency like worse spatial-temporal consistency between frames in video generation.

In particular, we propose to condition diffusion models on long-context such as text-prompt and past frames related to the fields. Text-prompt can represent data in compact but highly expressive features (Devlin et al., 2019; Brown et al., 2020; Raffel et al., 2020), and serve as a low-rank approximation of data (Radford et al., 2021). Past frames are especially useful in autoregressive generation, such as in game data. By conditioning diffusion models on long-context, we demonstrate that our method can capture the global geometry for generating long videos and game sequences.

**Text-prompt for Cross-view Condition Consistency.** In order to model the dependency variation between views belonging to the same field, *i.e.*, the global geometry of the field, we condition the diffusion model on the text embeddings of the field description or equivalent embeddings (*i.e.*, the language embedding of a single view in the CLIP latent space (Radford et al., 2021)). Our approach leverages the adaptive layer normalization layers in GANs (Brock et al., 2019; Karras et al., 2019), and adapts them by modeling the statistics from the text embeddings of shape $Z \times D$. For pairs that make up a single view, we condition on their represented tokens $Z \times D$, ($Z$ tokens of size $D$), by modulating them with the scale and shift parameters regressed from the text embeddings. For pairs $(T \times Z) \times D$ that make up multiple views, we condition on the view-level pairs by modulating feature in $Z \times D$ for each of the $T$ views with the same scale and shift parameters. Specifically, each transformer blocks of our score network learns to predict statistic features $\beta_c$ and $\gamma_c$ from the

text embeddings per channel. These statistic features then modulate the transformer features $F_c$ as: $\text{adLNorm}(F_c | \beta_c, \gamma_c) = \text{Norm}(F_c) \cdot \beta_c + \beta_c$.

**Past frames for Autoregressive Generation.** Recent work (Mei & Patel, 2023) shown that generating long videos and games can be formulated as autoregressive generation, where each frame depends only on past frames and current actions. In Figure. 3, we illustrate the input and output of our model, where it conditions on the past $T$ frames and generates the next $T + 1$ frame. Additional inputs include coordinates consisting of spatiotemporal coordinates and one-hot encoded actions, such as *jump* and *move*, from the last frame. Due to memory constraints, the input past frames are limited to a fixed number of $n$ past frames, acting as a sliding window for long-context modeling. The generation of the next $n$ frames on the diffusion field can be simplified as

$$p\left(\mathbf{y}_1, \mathbf{y}_2 \ldots, \mathbf{y}_{n-1}, \mathbf{y}_{(n,t-1)}\right) = \prod_{i=1}^{n} p\left(\mathbf{y}_{(n,t-1)} \mid \mathbf{y}_1, \mathbf{y}_2, \ldots, \mathbf{y}_{n-1}\right), \tag{1}$$

where $p(\cdot)$ represents the modeled signal probability conditioned on the past frames. Additional conditions also include coordinate inputs $(\mathbf{m}_1, \mathbf{m}_2, \ldots, \mathbf{m}_n)$ and the diffusion timestep $t$. Empirically, we use the last 16 frames as the context length for game generation, and the last 8 frames as the context length for text-to-video generation.

The proposed autoregressive generation not only preserves global geometry of the data but also significantly improves efficiency in long-context generation. Typical autoregressive transformer models like GPT (Radford et al., 2019) depend on the number of generated tokens, as each new token is conditioned on all previously generated tokens. In contrast to GPT, our method achieves linear complexity with respect to the number of generated frames, similar to the parallel generation efficiency. Each new frame depends only on a fixed number of the most recently generated frames, where conditioning frames are updated in the sliding window. Our game generation maximizes this efficiency, enabling the stable generation of games with an infinite number of frames.

## 4 EXPERIMENTAL RESULTS

We demonstrate the effectiveness of our method on multiple modalities, including 2D image data on a spatial metric space $\mathbb{R}^2$, 3D video data on a spatial-temporal metric space $\mathbb{R}^3$, and 3D viewpoint data on a camera pose and intrinsic parameter metric space $\mathbb{R}^6$, game data on a action and spatial-temporal metric space $\mathbb{R}^4$, while the score network implementation remains identical across different modalities, except for the embedding size.

**Experimental Details.** In the interest of maintaining simplicity, we adhere to the methodology outlined by Dhariwal et al. (Dhariwal & Nichol, 2021) and utilize a 256-dimensional frequency embedding to encapsulate input denoising timesteps. This embedding is then refined through a two-layer Multilayer Perceptron (MLP) with Swish (SiLU) activation functions. Our model aligns with the size configuration of DiT-XL (Peebles & Xie, 2023), which includes retaining the number of transformer blocks (*i.e.* 28), the hidden dimension size of each transformer block (*i.e.*, 1152), and the number of attention heads (*i.e.*, 16). Our model derives text embeddings employing T5-XXL (Raffel et al., 2020), culminating in a fixed length token sequence (*i.e.*, 256) which matches the length of the noisy tokens. To further process each text embedding token, our model compresses them via a single layer MLP, which has a hidden dimension size identical to that of the transformer block. Our model uses classifier-free guidance in the backward process with a fixed scale of 8.5. To keep consistency with DiT-XL (Peebles & Xie, 2023), we only applied guidance to the first three channels of each denoised token.

**Generative Metrics.** In video generation, we use FVD (Unterthiner et al., 2018) to evaluate the video spatial-temporal coherency, FID (Heusel et al., 2017) to evaluate the frame quality, and CLIP-SIM (Radford et al., 2021) to evaluate relevance between the generated video and input text. As all metrics are sensitive to data scale during testing, we randomly select 2,048 videos from the test data and generate results as the "real" and "fake" part in our metric experiments. For FID, we uniformly sample 4 frames from each video and use a total of 8,192 images. For CLIPSIM, we calculate the average score across all frames. We use the "openai/clip-vit-large-patch14" model for extracting features in CLIPSIM calculation.

| Model | CIFAR10 64×64 | | CelebV-Text 256×256×128 | | | ShapeNet-Cars 128×128×251 | | | |
|---|---|---|---|---|---|---|---|---|---|
| | FID (↓) | IS (↑) | FVD (↓) | FID (↓) | CLIPSIM (↑) | FID (↓) | LPIPS (↓) | PSNR (↑) | SSIM (↑) |
| Functa (Dupont et al., 2022a) | 31.56 | 8.12 | ✗ | ✗ | ✗ | 80.30 | 0.183 | N/A | N/A |
| GEM (Du et al., 2021) | 23.83 | 8.36 | ✗ | ✗ | ✗ | ✗ | ✗ | ✗ | ✗ |
| DPF (Zhuang et al., 2023) | 15.10 | 8.43 | ✗ | ✗ | ✗ | 43.83 | 0.158 | 18.6 | 0.81 |
| DiT (Peebles & Xie, 2023) | 7.53 | 8.97 | ✗ | ✗ | ✗ | | ✗ | ✗ | ✗ |
| TFGAN (Balaji et al., 2019) | ✗ | ✗ | 571.34 | 784.93 | 0.154 | ✗ | ✗ | ✗ | ✗ |
| MMVID (Han et al., 2022b) | ✗ | ✗ | 109.25 | 82.55 | 0.174 | ✗ | ✗ | ✗ | ✗ |
| MMVID-interp (Han et al., 2022b) | ✗ | ✗ | 80.81 | 70.88 | 0.176 | ✗ | ✗ | ✗ | ✗ |
| VDM (Ho et al., 2022b) | ✗ | ✗ | 81.44 | 90.28 | 0.162 | ✗ | ✗ | ✗ | ✗ |
| CogVideo (Hong et al., 2023) | ✗ | ✗ | 99.28 | 54.05 | 0.186 | ✗ | ✗ | ✗ | ✗ |
| Latte (Ma et al., 2024) | ✗ | ✗ | 67.97 | 39.69 | 0.201 | ✗ | ✗ | ✗ | ✗ |
| EG3D-PTI (Chan et al., 2022) | ✗ | ✗ | ✗ | ✗ | ✗ | 20.82 | 0.146 | 19.0 | 0.85 |
| ViewFormer (Kulhánek et al., 2022) | ✗ | ✗ | ✗ | ✗ | ✗ | 27.23 | 0.150 | 19.0 | 0.83 |
| pixelNeRF (Yu et al., 2021) | ✗ | ✗ | ✗ | ✗ | ✗ | 65.83 | 0.146 | 23.2 | 0.90 |
| Zero-1-to-3 (Liu et al., 2023) | ✗ | ✗ | ✗ | ✗ | ✗ | **17.901** | **0.093** | 23.1 | 0.80 |
| **Field-DiT (Ours)** | **7.29** | **9.31** | **42.03** | **24.33** | **0.220** | 24.36 | 0.118 | **23.9** | **0.90** |

Table 1: Sample quality comparison with state-of-the-art field models and representative modality-specific models for each task. "✗" denotes that the method cannot be applied to the modality due to its design or impractical computational costs.

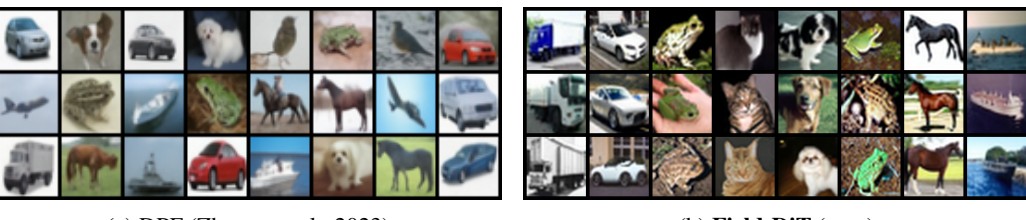

(a) DPF (Zhuang et al., 2023)        (b) **Field-DiT** (ours)

Figure 4: Qualitative comparisons of domain-agnostic methods and ours on CIFAR-10. Our results show better visual quality with more details than the others, while being domain-agnostic as well.

**Images.** For image generation, we use the standard benchmark dataset, *i.e.*, CIFAR10 64×64 (Krizhevsky et al., 2009) as a sanity test, in order to compare with other domain-agnostic and domain-specific methods. For the low-resolution CIFAR10 dataset, we compare our method with the previous domain-agnostic methods including DPF (Zhuang et al., 2023) and GEM (Du et al., 2021). We report Fréchet Inception Distance (FID) Heusel et al. (2017) and Inception Score (IS) (Salimans et al., 2016) or quantitative comparisons.

The experimental results can be found in Tab. 1. Specifically, Field-DiT outperforms all domain-agnostic models in the FID and IS metrics. The qualitative comparisons in Fig. 4 further demonstrate our method's superiority in images. Note that our method does not use text descriptions for ensuring a fair comparison. It simply learns to predict all coordinate-signal pairs of a single image during training without using additional text descriptions or embeddings.

**Videos.** To show our model's capacity for more complex data, *i.e.*, high-resolution, dynamic video, we conduct experiments on the recent text-to-video benchmark: CelebV-Text 256×256×128 (Yu et al., 2023b) (128 frames). As additional spatial and temporal coherence is enforced compared to images, video generation is relatively underexplored by domain-agnostic methods. We compare our method with the representative domain-specific methods including TFGAN (Balaji et al., 2019), MMVID (Han et al., 2022a), CogVideo (Hong et al., 2023), VDM (Ho et al., 2022b), and Latte (Ma et al., 2024). We report Fréchet Video Distance (FVD) (Unterthiner et al., 2018), FID, and CLIP-SIM (Wu et al., 2021), *i.e.*, the cosine similarity between the CLIP embeddings (Radford et al., 2021) of the generated images and the corresponding texts.

Our method achieves the comparable performance in both the video quality (FVD) and signal frame quality (FID) in Tab. 1, compared with the recent domain-specific text-to-video models. Moreover, our model learns more semantics as suggested by the CLIPSIM scores. The results show that our model, as a domain-*agnostic* method, can achieve a performance on par with domain-*specific* methods in modeling long-context. The qualitative comparisons in Fig. 5 further support our model in text-to-video generation compared with the recent state-of-the-art methods.

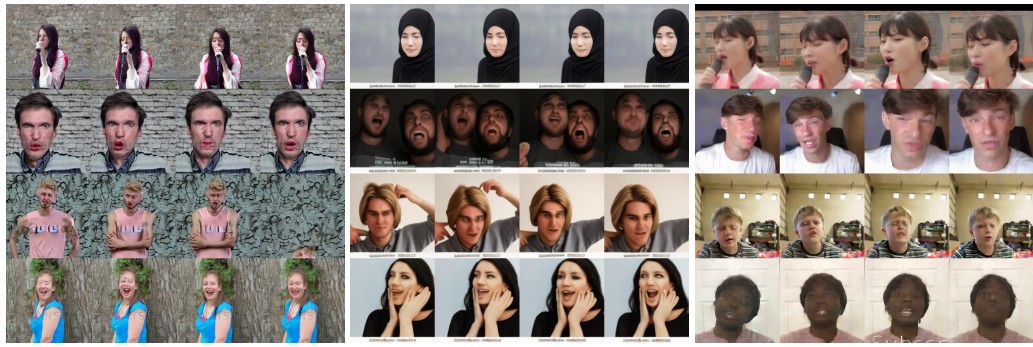

(a) VDM (Ho et al., 2022b)    (b) CogVideo (Hong et al., 2023)    (c) **Field-DiT** (Ours)

Figure 5: Qualitative comparisons between domain-specific text-to-video models and ours. Compared to VDM Ho et al. (2022b), our results are more continuous. Compared to CogVideo Hong et al. (2023), our results feature more realistic textures. Please see `https://kfmei.com/Field-DiT/` for the input prompt and video results.

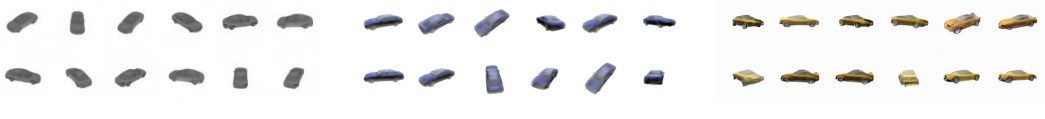

(a) pixelNeRF (Yu et al., 2021)    (b) Functa (Dupont et al., 2022b)    (c) **Field-DiT** (Ours)

Figure 6: Qualitative comparisons between representative 3D novel view generation methods and ours. Our results demonstrate competitive quality without explicitly using 3D modeling.

| PSNR (dB) | 10% | 20% | 30% | 40% | 50% | 60% | 70% | 80% | 90% | 100% |
|---|---|---|---|---|---|---|---|---|---|---|
| DPF (Zhuang et al., 2023) | 24.00 | 21.97 | 20.87 | 20.66 | ✗ | ✗ | ✗ | ✗ | ✗ | ✗ |
| **Field-DiT** (Ours) | 44.30 | 43.96 | 43.87 | 44.16 | 42.92 | 42.20 | 42.42 | 42.51 | 42.07 | 42.22 |

Table 2: We demonstrate the long-context modeling capability of our model by showing its next-frame generation accuracy on game data, where a total of 100 frames are evaluated. ✗ denotes out-of-memory results when the model cannot handle such a long context.

**3D novel views.** We also evaluate our method on 3D novel view generation with the ShapeNet dataset (Chang et al., 2015). Specifically, we use the "car" class of ShapeNet which involves 3514 different cars. Each car object has 50 random viewpoints, where each viewpoint is in $128 \times 128$ resolution. Unlike previous domain-agnostic methods (Du et al., 2021; Zhuang et al., 2023) that model 3D geometry over voxel grids at $64^3$ resolution, we model over rendered camera views based on their corresponding camera poses and intrinsic parameters, similar to recent domain-specific methods (Sitzmann et al., 2019; Yu et al., 2021). This approach allows us to extract more view-wise coordinate-signal pairs while voxel grids only have 6 views. We report our results in comparison with the state-of-the-art view-synthesis algorithms including pixelNeRF (Yu et al., 2021), viewFormer (Kulhánek et al., 2022), EG3D-PTI (Chan et al., 2022), and Zero-1-to-3 Liu et al. (2023). Among the compared methods, only Zero-1-to-3 is a zero-shot generation approach, while the others are trained on the ShapeNet dataset, either by their authors or by us. Zero-1-to-3 is pretrained on large-scale data, making it robust to out-of-domain data. Note that our model performs one-shot novel view synthesis by conditioning on the text embedding of a random view. Compared to recent methods specifically designed for 3D modalities, our approach achieves higher fidelity metrics, such as PSNR and SSIM, while producing comparable scores in LPIPS. Although methods like EG3D-PTI and Zero-1-to-3, which directly fine-tune pretrained 2D image generation models like StyleGAN and Stable-Diffusion, achieve better FID scores, this metric prioritizes 2D visual quality. However, it does not strictly reflect 3D consistency, which limits its relevance for 3D evaluation.

Concurrent 3D novel view generation methods, such as Eschernet (Kong et al., 2024), Wonder3D (Long et al., 2024), and SV3D (Voleti et al., 2025), have demonstrated their effectiveness in modeling sparse views of 3D objects. While our method may underperform in perceptual quality due to its modality-unified design, its unique ability to generate continuous, long-context 3D views offers a fresh perspective to the 3D modeling community.

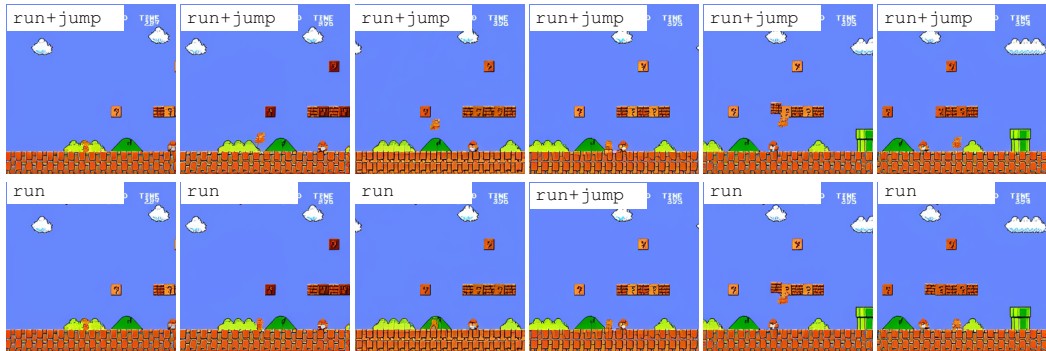

Figure 7: Visualization of our generated game (1/8 sampling rate at 50 frames), showcasing how our method generalizes to different actions within the same context. Each frame's action is labeled in the top-left corner. Please see `https://kfmei.com/Field-DiT/` for videos.

| Text | Cross-view consistent noise | Resolution | Training Views $n$ | FVD ($\downarrow$) | FID ($\downarrow$) | CLIPSIM ($\uparrow$) | MACs | Mems |
|------|------|------|------|------|------|------|------|------|
| ✗ | N/A | 16.0 | 8 | 608.27 | 34.10 | - | 113.31G | 15.34Gb |
| ✓ | ✗ | 16.0 | 8 | 401.64 | 75.81 | 0.198 | 117.06G | 15.34Gb |
| ✓ | ✓ | 1.0 | 8 | 115.20 | 40.34 | 0.187 | 7.314T | 22.99Gb |
| ✓ | ✓ | 16.0 | 1 | 320.02 | 21.27 | 0.194 | 117.06G | 15.34Gb |
| ✓ | ✓ | 16.0 | 4 | 89.83 | 23.69 | 0.194 | 117.06G | 15.34Gb |
| ✓ | ✓ | 16.0 | 8 | 42.03 | 24.33 | 0.220 | 117.06G | 15.34Gb |

Table 3: Ablation analysis of the text-to-video results of our proposed method under different settings. All computation costs (MACs) and GPU memory usage (Mems) are estimated for generating a single view, regardless of the resolution, to ensure a fair comparison. The mark in the text column indicates whether a text prompt is used. The number in the resolution column denotes the usage of a latent encoder, where a resolution equal to 1 means the model is directly trained in pixel space.

**Games.** Game generation is an under-explored area and lacks data and benchmarks. We demonstrate the game generation capability of our method by showing the accuracy of predicted frames compared with the frame of the real game when using the same action. Specially, we model the World 1-1 of Super Mario Bros (NES version) with a sliding window size of 16, and we test it with new actions for next-frame generation. Fig. 7 shows the visual results generated from two different actions starting from the same scene. Tab. 2 demonstrates our long-context modeling capability compared with the DPF, where ours performance loss is minor compared with DPF.

## 4.1 ABLATIONS AND DISCUSSIONS

In this section, we demonstrate the effectiveness of each of our proposed components and analyze their contributions to the quality of the final result, as well as the computation cost. The quantitative text-to-video generation results under various settings are shown in Table 3.

**Effect of text condition.** To verify the effectiveness of the text condition for capturing the global geometry of the data, we use two additional settings. **(1)** The performance of our model when the text condition is removed is shown in the first row of Tab. 3. The worse FVD means that the text condition play a crucial role in preserving the global geometry, specifically the spatial-temporal coherence in videos. **(2)** When the text condition is added, but not the cross-view consistent noise, the results can be found in the second row of Tab. 3. The FVD is slightly improved compared to the previous setting, but the FID is weakened due to underfitting against cross-view inconsistent noises. In contrast to our default setting, these results demonstrate the effectiveness of the view-consistent noise. Furthermore, we note that more detailed text descriptions can significantly improve the generated video quality.

**Effect of number of views.** We investigate the model performance change with varying number of views ($n$) for representing fields, as shown in the 2nd and 3rd rows of Tab. 3. Compared to the default setting of $n = 8$, reducing $n$ to 1 leads to non-continuous frames and abrupt identity changes, as indicated by the low FVD. When $n$ is increased to $4$, the continuity between frames is improved, but still worse than $n = 8$ for the dynamics between frames. Thus, we can conclude that a larger number of views leads to a higher performance, along with a higher computation cost.

**Comparison with Context Query Pairs.** Even though context query pairs introduced by DPF (Zhuang et al., 2023) has been justified to achieve better performance than using latent space (which needs reconstruction training) in small models and low-resolution modalities, it is shown (Zhuang et al., 2023) to be impossible to largely reduce the memory footprint (by sampling less context pairs) while preserving its original modeling capability and performance. To scale up our model, we replace the context query pairs with latent space in our method. It can significantly reduce memory usage (*e.g.* using less than 2% pairs while maintaining a competitive performance) so that handling a larger model size becomes possible with high-resolution, long views. Based on these, the benefit of scaling using the latent space outweighs the potential performance loss led by the latent space, as backed by Tab. 1.

**Comparison with Modality Unified Models.** Our method shares the motivation of modality-unified models like SIREN and Functa for handling diverse data modalities but differs in complexity and scope. SIREN uses sinusoidal activations in MLPs to represent continuous signals, excelling in modeling structured data and solving mathematical problems like PDEs with high fidelity but is limited to simpler datasets due to its MLP architecture. In contrast, our diffusion transformer framework handles more diverse and complex data, integrating view-wise sampling for local structure and autoregressive generation for global consistency. Additionally, text and past frame conditioning enable Field-DiT to scale effectively to complex multi-modal tasks, making it more versatile for dynamic and high-dimensional datasets compared to SIREN's structured focus.

## 5 LIMITATIONS.

(1) Our method only applies to visual modalities interpretable by views. For modalities such as temperature manifold Hersbach et al. (2019) where there is no "views" of such field, our method does not apply. As long as the data in the new domain (e.g., 3D dynamic scene and MRI) can be interpreted by views, our method can reuse the same latent autoencoder Rombach et al. (2022) without switching to domain-specific autoencoders. (2) Our method aligns with the standard practice outlined in DPF, using comparison methods with weights trained separately for each modality. While it performs exceptionally well on individual modalities, achieving strong performance across multiple modalities simultaneously is hindered by the inherent challenges of the multi-task problem. We believe our approach provides a solid foundation for the future versatile generative models.

## 6 CONCLUSION

We have introduced a new transformer-based diffusion field model that addresses the limitations of current probabilistic field models in capturing global structures and long-context dependencies. By utilizing a view-wise sampling algorithm for local structure learning and incorporating autoregressive generation to preserve global geometry, our approach overcomes the shortcomings of MLP-based architectures. The proposed model can generate high-fidelity data across multiple modalities, including text-to-video, 3D view generation, and game control while maintaining scalability and unifying diverse modalities.

**Acknowledgments.** Vishal M. Patel was supported by NSF CAREER award 2045489.

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
