# OpenReview forum: "Field-DiT: Diffusion Transformer on Unified Video, 3D, and Game Field Generation"
_ICLR.cc/2025/Conference — ICLR 2025 Poster_

### Official Review · Reviewer_x2pz · 2024-10-31

**Soundness:** 2
**Presentation:** 1
**Contribution:** 2
**Rating:** 3
**Confidence:** 4

**Summary:**

This paper focuses on unifying data generation across various modalities, including images, videos and 3D geometry. It introduces a view-wise sampling algorithm along with autoregressive generation to improve the performance. The proposed framework can handle various modalities with good performance.

**Strengths:**

1. The proposed method is a unified framework for various modalities. This is a relatively new task that might benefit the community.

2. Extensive experiments and ablation studies demonstrate the effectiveness of the proposed method.

**Weaknesses:**

1. **The motivation is unclear.** From the introduction section, the main motivation to use Diffusion Probabilistic Field (DPF) is handling various modalities together with a unified model. As for the unified models, what I understand is a single model that can generate different modalities. However, from the description in the method and experiment sections, each modality has different coordinate-signal pairs and the models are trained for each modality separately. If so, such a method cannot be regarded as a unified framework in my view.

2. **Comparison with conventional diffusion models.** In Line 142-144, when comparing DPF with conventional diffusion models, the main difference is that DPF can be applied to sparse observation of fields. However, in the view-wise sampling subsection (Line 244), each time sample the tokens in n views, which is a dense modeling instead of sparse sampling.

3. **Comparison with DPF.** In my view, the main contribution of DPF is the context query pairs sampling and optimization. However, in Line 502, this paper mentions that the context query pairs are not used, which confuses me about the training objective in this paper. Does this paper use the diffusion optimization objective like epsilon-prediction or velocity-prediction? If so, the method is almost the same with DiT.

4. **Limited performance.** I do not see a part describing the dataset and hyperparameters used for training. So I assume the model is trained on each benchmark. If so, the performance is far from satisfactory since the compared methods are generalizable ones instead of fitting to a benchmark.

**Questions:**

I am not an expert in Diffusion Probabilistic Fields, and the writing of this paper makes me even more confusing. I hope the authors could improve the writing and explain more background and related work. In addition, most of my concerns are about the explanation of the method and motivation. Please refer to weaknesses for more details.

## Post Rebuttal
After discussing with the authors, I have decided to assign a reject score to this paper. Below are the key concerns that remain unaddressed:

- **Limited Contribution of the Unified Architecture**: The contribution of a unified architecture trained separately for each dataset is limited, especially since a similar concept has already been introduced by DPF.
- **In-Distribution vs. Out-of-Distribution Performance**: For a generalizable model, the in-distribution performance should surpass out-of-distribution performance. For instance, in the comparison with Zero-1-to-3 in 3D generation, this expectation is not met.
- **Video Generation Quality**: The qualitative results for video generation on the rebuttal page are unsatisfactory. None of the examples demonstrate temporally consistent motion. In contrast, the webpage for the compared method, [Latte](https://maxin-cn.github.io/latte_project/), showcases much better video generation results.
- **Game Generation Quality**: The qualitative results for game generation are also unsatisfactory. Comparing the author-provided [game recording](https://www.youtube.com/watch?v=Gum4GI2Jr0s) with the results on the rebuttal page, the color of the bricks at the bottom of the generated videos is inconsistent. Furthermore, the generated scenes are limited, lacking elements such as flowers and turtles present in the recording, as well as any underground scenarios.

---

> ### Author Response · Authors · 2024-11-17
> **Response by Authors (Nov 17)**
>
> We thank the reviewer for the valuable feedback. We address all concerns below, and revise the manuscript accordingly.
>
> > W1. The motivation is unclear. From the introduction section, the main motivation to use Diffusion Probabilistic Field (DPF) is handling various modalities together with a unified model. As for the unified models, what I understand is a single model that can generate different modalities. However, from the description in the method and experiment sections, each modality has different coordinate-signal pairs and the models are trained for each modality separately. If so, such a method cannot be regarded as a unified framework in my view.
>
> The unification lies not in training a single model weight for all modalities simultaneously but in employing a single, versatile architecture that can be applied across different modalities without modification.
>
> The primary motivation of recent unified field models like DPF is to develop a framework that models multiple modalities through the same network architecture and training paradigm. Such a unified architecture simplifies the development process, reduces the complexity associated with designing modality-specific models, and enables knowledge transfer between modalities.
>
> We acknowledge that training a single model with weights capable of handling different modalities is useful. However, ensuring that the model performs well on each modality simultaneously is a multi-task learning problem. Addressing this challenge differs from our primary focus on long-context modeling. We consider this an exciting avenue for future research. Therefore, like DPF, our method uses the same network architecture but trains separate sets of weights for each modality. Our experiment also follows the DPF evaluation protocol, which uses separate model weights per modality, ensuring fair comparisons.
>
> The revised introduction section clarifies the motivation of field models to demonstrate the aforementioned content.
>
> ---
>
> > W2. Comparison with conventional diffusion models. In Line 142-144, when comparing DPF with conventional diffusion models, the main difference is that DPF can be applied to sparse observation of fields. However, in the view-wise sampling subsection (Line 244), each time sample the tokens in n views, which is a dense modeling instead of sparse sampling.
>
> At a high level, the key difference among our work, DPF, and conventional diffusion models lies mainly the methods lies in the degree of sparsity during sampling:
> - **Conventional Diffusion Models**: Operate on the complete data, sampling all data points (e.g., every frame in a video) simultaneously. This is dense modeling with no sparsity.
> - **DPF**: Samples a very small subset of the data, such as individual pixels, representing the highest level of sparsity.
> - **Our Method**: Introduces an intermediate level of sparsity by sampling a small subset of n views (e.g. 8) out of a large set of N views (e.g. 128), where n≪N.
>
> Sampling n views might appear dense. However, it represents a sparsity level between processing the complete data and DPF's pixel-level sampling.
>
>
> Specifically, compared to DPF, our method sacrifices some flexibility (we focus on specific data types like images and videos rather than diverse modalities like scientific temperature fields). However, we gain improved performance and visual quality by preserving local structures within views. Compared to Conventional diffusion models, we reduce computational requirements by not processing the  complete data at once. We also maintain higher performance than extremely sparse methods due to our intermediate level of sparsity. Therefore, our method lies between dense modeling and sparse modeling, achieving a balance between the high flexibility (but lower performance) of DPF and the high performance (but lower flexibility) of conventional diffusion models.
>
> The revised related work clarifies the difference in sparsity and demonstrates the aforementioned content.

---

> ### Author Response · Authors · 2024-11-17
> **More Response by Authors (Nov 17)**
>
> > W3. Comparison with DPF. In my view, the main contribution of DPF is the context query pairs sampling and optimization. However, in Line 502, this paper mentions that the context query pairs are not used, which confuses me about the training objective in this paper. Does this paper use the diffusion optimization objective like epsilon-prediction or velocity-prediction? If so, the method is almost the same with DiT.
>
> **Clarification**: Our method uses velocity-prediction, which is a standard diffusion loss. We would like to clarify that modality-unified methods like DPF don’t have a special loss function; they also use the standard diffusion loss.
>
> **Response**: Our method is not based on DPF. Our method shares a similar motivation with DPF for modality-unified modeling. However, our method is significantly different from DPF, primarily in the degree of sparsity during sampling, as we responded to the second concern. Our method adopts the DiT network architecture and modifies its sampling paradigm into sparse representation to accommodate video, 3D, and game generation, as DiT is limited to generating 2D images. The global geometry of sampled results is enhanced by two new mechanisms including cross-view consistent conditions and autoregressive generation. Table 3 in our ablation study also demonstrates the results of directly using DiT to model complex video data. This results in either significantly higher computational costs (7.314T vs. 117.06G) or markedly worse performance (608 FVD vs. 42 FVD) compared to our improved version.
>
>
> We don’t use context-query pairs proposed by DPF  because context-query is only useful for enhancing the global context during sparse sampling, however, its help is not significant as shown in the Table 5 of DPF’s paper, where its best performance can only be achieved when using 100% of samples as the context-pairs. Such a number is essentially equivalent to modeling the whole data as a single data point, which has the same memory and computation as the conventional diffusion model. Therefore, our method reduces the burden of using context-query pairs by introducing a new view-wise sampling mechanism. The global context during sampling is further strengthened by introducing two strategies, including cross-view consistent conditions and autoregressive generation. The ablation in Table 3 comprehensively demonstrates the effectiveness of each module.
>
>
> The revised related work clarifies the difference with DiT. Line 503-5123 demonstrates the aforementioned content in detail.
>
>
> ---
>
> > W4. Limited performance. I do not see a part describing the dataset and hyperparameters used for training. So I assume the model is trained on each benchmark. If so, the performance is far from satisfactory since the compared methods are generalizable ones instead of fitting to a benchmark.
>
> We clarify the evaluation protocol in response to your W1 inquiry. Specifically, we evaluate our method on various benchmarks using distinct model weights, without combining multiple modalities into a single model as is done in multi-task learning. Consequently, we do not have hyper-parameters for balancing modality contributions to the learning process. This evaluation protocol is consistent with the one used by our modality-unified baseline, ensuring an apples-to-apples comparison with other unified-modality methods.
>
> ---
>
> We rewrite our related work in the reversion to thoroughly explain the difference in motivation between unified-modality modeling and our method. We hope that the improved writing and presentation will help the reviewer understand the background of the problem and the details of our method more clearly.

---

> ### Author Response · Authors · 2024-11-22
> **A Kind Discussion Deadline (11/26) Reminder**
>
> We have addressed the reviewer's concerns, including clarifying the motivation, highlighting the differences with conventional diffusion models and DPF, and addressing the perceived "limited performance." As the discussion deadline (November 26th) is approaching, we wanted to follow up and see if you've had a chance to review our response.  We understand you may be facing a tight deadline for submitting CVPR papers. We hope everything is going smoothly.
>
> We sincerely hope our responses have addressed your concerns.  Please let us know if you have any further questions; we are happy to provide additional clarification or experimental results.

---

> > ### Comment · Area_Chair_gcFn · 2024-11-24
> > **Discussion Period Ending Soon**
> >
> > Dear Reviewer x2pz,
> >
> > The discussion period will end soon. Please take a look at the author's comments and begin a discussion.
> >
> > Thanks, Your AC

---

> ### Author Response · Authors · 2024-11-26
> **Follow-Up on Paper #824**
>
> Dear Reviewer x2pz,
>
> We hope this message finds you well and at a convenient time. Based on the timing of your previous review submission, we believe this moment aligns with your schedule and is an appropriate opportunity to follow up.
>
> We have carefully addressed the concerns you raised and revised the manuscript accordingly. However, we have not yet received further feedback and would like to confirm that we have fully addressed all your concerns. If any points remain unclear or are preventing a more positive evaluation, please let us know, and we will gladly make further revisions as needed. Your valuable insights are greatly appreciated, and we sincerely hope to hear from you soon.
>
> Thank you for your time and understanding.
>
> Sincerely,
>
> The Authors of Paper #824

---

> > ### Comment · Reviewer_x2pz · 2024-11-26
> >
> > Thank you to the authors for their detailed response.
> >
> > Based on their explanation, I understand that the "unified model" described in the paper refers to a unified architecture for handling different tasks, although each task is trained separately. While I respect this interpretation of the problem definition, I do not fully agree with it.
> >
> > According to the paper's definition, each model is trained on the evaluation dataset, yet the performance remains unsatisfactory. For instance, in the case of 3D generation, the compared method, Zero-1-to-3, also generates one view at a time, similar to the training approach in this paper. However, Zero-1-to-3 is trained on a generalized dataset rather than the evaluation dataset (ShapeNet), and its performance significantly surpasses that of the proposed method.
> >
> > Additionally, one scenario where the so-called sparse sampling approach might be particularly useful is in generating long videos, such as sequences exceeding 100 frames, rather than comparing 8-frame versus 16-frame outputs. If sparse sampling were effective, the model should be capable of producing temporally consistent long videos. However, based on the results showcased on the webpage, the video generation performance appears to be quite poor.
> >
> > As a result, I have decided to lower my rating. I am open to further discussion in case there are any misunderstandings.

---

> ### Author Response · Authors · 2024-11-27
> **Follow-up Response (1/3) to Unified Model (Nov 26)**
>
> We thank the reviewer for the continued engagement and for providing valuable feedback. We address all concerns below, and revise the manuscript accordingly.
>
> **Ambiguous Modal-Unification Claim**: To enhance clarity, we revised the terminology from "unified models" to "architecture-unified models" throughout the manuscript. For example:
>
> - L#023: “highlight its potential as a foundational framework for scalable, modality-unified visual content generation.” -> “highlight its potential as a foundational framework for architecture-unified visual content generation for different modalities with different weights.”
>
> - L#100: “demonstrate … modalities unified generation including video, 3D, and game,” -> “demonstrate … different modalities generation including video, 3D, and game in a unified architecture with different weights,”
>
> Our future work section in revision discusses the advantages and disadvantages of training distinct weights for various modalities within the unified architecture used by our method. We believe our current method serves as a strong foundation for future modality-unified generation approaches.

---

> ### Author Response · Authors · 2024-11-27
> **Follow-up Response (2/3) to 3D Generation (Nov 26)**
>
> **Ineffective Comparison when Training on ShapeNet and Testing on ShapeNet**: To enhance clarity, our revision in L#427-428 highlights that the compared methods include EG3D-PTI (Chan et al., 2022), ViewFormer (Kulh´anek et al., 2022), and pixelNeRF (Yu et al., 2021) are trained on the ShapeNet training dataset, either by their authors or by us, similar as our method, ensuring a fair comparison. Compared to these methods, our method achieves the best performance on the ShapeNet testing dataset across all metrics: FID, LPIPS, PSNR, and SSIM.
>
> We acknowledge the potential unfairness in comparing the Zero-1-to-3 model without retraining it. However, as the model has already been trained on 800k diverse 3D models and demonstrates robustness when handling out-of-domain data, we decided not to retrain it due to computational constraints.
>
> **Inferior FID and LPIPS Compared with Zero-1-to-3**: In response to reviewer 4gwr’s concern, we noted: In our revision, L#431-458 discuss the result differences between 3D modeling and video generation. Specifically, during 3D evaluation, there is a Perception-Distortion Tradeoff (Blau and Michaeli 2018). The perceptual quality is reflected by the FID and LPIPS metrics, and the 3D distortion loss is reflected by the PSNR and SSIM metrics. 3D-specific models such as Zero-1-to-3 and EG3D-PTI, fine-tuned from StableDiffusion or StyleGAN, prioritize perceptual quality, achieving higher FID and LPIPS scores. In contrast, our method, trained from scratch, leads to an emphasis on 3D reconstruction consistency, empirically achieving better PSNR and SSIM scores.
>
> **Advantages over Zero-1-to-3 Due to Unified Architecture**: Our method is capable of producing continuous, long-context 3D novel views by leveraging long-context modeling capabilities. In contrast, Zero-1-to-3 generates discrete views, which limits the smoothness and temporal consistency of the output.
>
> To demonstrate this, we provide additional visual comparisons. Note that our method is trained on the car category of ShapeNet and tested on the chair category of ShapeNet, ensuring better fairness in the evaluation between our method and Zero-1-to-3.
>
> - Chair1: ours (128 frames, takes 1.5 minutes to generate)  https://transdif-web.pages.dev/visual-comparisons-shapenet/ours_000000.mp4  vs. Zero-1-to-3 (128 frames, takes 5 minutes to generate) https://transdif-web.pages.dev/zero_1_to_3_chair_1.mp4
>
>
> - Chair2: ours (128 frames, takes 1.5 minutes to generate)  https://transdif-web.pages.dev/visual-comparisons-shapenet/ours_000013.mp4  vs. Zero-1-to-3 (128 frames, takes 5 minutes to generate) https://transdif-web.pages.dev/zero_1_to_3_chair_2.mp4
>
>
> - Chair3: ours (128 frames, takes 1.5 minutes to generate)  https://transdif-web.pages.dev/visual-comparisons-shapenet/ours_000009.mp4  vs. Zero-1-to-3 (128 frames, takes 5 minutes to generate) https://transdif-web.pages.dev/zero_1_to_3_chair_3.mp4
>
>
> The ability to generate continuous sequences is particularly advantageous for precise 3D novel view generation.

---

> ### Author Response · Authors · 2024-11-27
> **Follow-up Response (3/3) to Video Generation (Nov 26)**
>
> **Quantitative Video Results Beyond 8 and 16 Frames**: In our paper, Table 1 and Table 2 present quantitative results for long-sequence generation: 128 frames at 256×256 resolution for videos and 100 frames at 256×256 resolution for game simulations. Our method achieves the best FVD score of 42.04 for videos and a PSNR of 42.22 for games.
>
> **Qualitative Video Results Beyond 8 and 16 Frames**: Our supplementary materials, available at https://transdif-web.pages.dev/, include visualizations of 12-second game sequences and 7-second videos. These examples demonstrate our model's ability to generate temporally consistent long sequences.
>
> **Additional 640 Frames Long-context Generation**: Our method is capable of generating frames of infinite length while maintaining consistent spatial-temporal coherence. Since the 128-frame length of our training videos limits the output in text-to-video generation, we present game generation results below to demonstrate the long-context generation capability.
> - Clip1: 640 frames at 8 FPS, resulting in a 2-minute-long game simulation  https://transdif-web.pages.dev/long_game_640_frames_2min.mp4
>
> **Inferior Aesthetic Quality**: We acknowledge the inferior aesthetic quality resulting from our current method being a single-stage generative model. State-of-the-art (SOTA) video-specific methods, such as Stable Video Diffusion (Blattmann et al., 2023), employ an additional stage for high-quality video refinement. However, creating such a supervised fine-tuning (SFT) model typically requires access to commercial text-to-video data and costly data curation, which are unavailable to us. As aesthetic quality has minimal impact on demonstrating spatial-temporal consistency, we have chosen not to pursue SFT training and leave it as future work by using curated data.
>
> The superior quantitative performance  in Table 1 is achieved without relying on specialized modules such as optical flow or temporal attention, thereby highlighting the effectiveness of our sparse sampling approach.
>
> ---
>
> We appreciate the reviewer's insights and hope that our explanations address the concerns raised. If any points remain unclear or are preventing a more positive evaluation, please let us know, and we will gladly make further revisions as needed.

---

> ### Author Response · Authors · 2024-11-27
> **[Last day of paper revision] We are anticipating your feedback!**
>
> Dear Reviewer x2pz,
>
> It's the last day that we are allowed to revise the paper. We would be so grateful if you could kindly check our responses and revised paper and let us know if you are happy with the improvement we've made according to your advice.
> Please do not hesitate to reach out if you have any further questions or require additional clarifications. We are more than happy to have the valuable chance to continue the discussion with you.
>
> Thank you very much for your time and consideration.
>
> Sincerely,
>
> Authors of Paper A Simple Diffusion Transformer on Unified Video, 3D, and Game Field Generation

---

> > ### Comment · Reviewer_x2pz · 2024-11-27
> >
> > Thank  authors for the response. However, my biggest concern regarding the performance has not been addressed.
> >
> > For the 3D generation, I agree that FID may not be an ideal evaluation metric for 3D generation, but since it is listed in the table, the poor performance on this metric should still be considered. PSNR, SSIM, and LPIPS are commonly used metrics for 3D generation. However, this paper achieves only comparable performance on PSNR and worse performance on LPIPS. Additionally, it is important to note that Zero-1-to-3 was published at ICCV 2023, and there are many new papers on 3D generation with better performance that should be compared. For example, EscherNet (Kong et al., 2024), Wonder3D (Long et al., 2024), and SV3D (Voleti et al., 2024). Most of the methods compared in this paper—EG3D-PTI (Chan et al., 2022), ViewFormer (Kulhánek et al., 2022), and pixelNeRF (Yu et al., 2021)—are outdated and are not generative models. Regarding the qualitative results provided, considering the quantitative results, the examples seem to be cherry-picked.
> >
> > For the video generation, based on the qualitative results provided on the rebuttal webpage, it is difficult to regard them as good video generation results. Both the quality and temporal consistency are unsatisfactory. Moreover, the 640-frame video also lacks temporal consistency; for instance, the color of the bricks at the bottom of the video keeps changing.
> >
> > Regarding the response to "Inferior Aesthetic Quality," I am confused about the SFT mentioned by the authors. The video generation model uses captions and videos in a self-supervised manner without additional labels. Furthermore, many non-commercial text-to-video datasets are available for research, such as OpenVid-1M, Panda-70M, and WebVid-10M. Therefore, acquiring data should not be considered a problem.
> >
> > Therefore, I keep my rating as reject.

---

> ### Author Response · Authors · 2024-11-27
> **Follow-up Response (Nov 27)**
>
> We thank the reviewer again for their continued engagement and valuable feedback. We address all concerns below and have revised the manuscript accordingly.
>
> **Effectiveness of the Experimental Results in Supporting the Motivation**: Our goal is to propose a network capable of sampling major modalities, including images, videos, 3D views, and game simulations, within a unified architecture. Modeling these modalities at higher resolutions (256×256) and on complex data (more than 100 frames) is simply impossible for our baseline method, DPF. As demonstrated in Table 1, our proposed method not only significantly outperforms previous modality-unified models but also surpasses most state-of-the-art modality-specific approaches. Its comparable performance achieved without using any modality-specific modules comprehensively demonstrates the superiority of our architecture-unified approach.
>
> To enhance clarity, we revised the claim from "outperforming modality-specific methods" to "achieving comparable performance with representative modality-specific methods". For example, we changed our contribution summary in L#099 to: "... largely closes the performance gap with modality-specific models."
>
> **Methodological Differences with Concurrent 3D-Specific Methods**: In our revision of L#428, we discuss the methodological differences between our approach and concurrent 3D-specific methods, including EscherNet (Kong et al., 2024), Wonder3D (Long et al., 2024), and SV3D (Voleti et al., 2024). Our unique advantages in generating continuous 3D views with long-context modeling capability still offers a fresh perspective to the 3D modeling community.
>
> **Additional Continuous 3D Generation Results**: Chair4: [Ours](https://transdif-web.pages.dev/visual-comparisons-shapenet/ours_000002.mp4) vs. [Zero-1-to-3](https://transdif-web.pages.dev/zero_1_to_3_chair_4.mp4), Chair5: [Ours](https://transdif-web.pages.dev/visual-comparisons-shapenet/ours_000006.mp4) vs. [Zero-1-to-3](https://transdif-web.pages.dev/zero_1_to_3_chair_6.mp4)
>
> **Flickering Results in Game Generation**: The flickering is an inherent behavior of the Super Mario game. For example, similar flickering can be observed in the gameplay [recording](https://www.youtube.com/watch?v=Gum4GI2Jr0s).
>
> **Inferior Training Data**: Our supplementary materials (available online at https://transdif-web.pages.dev/) note that our text-to-video generation results were trained on the WebVid dataset. The poor aesthetic quality of WebVid, as also highlighted by Ma et al. (2024), leads to inferior aesthetic quality in the generated results.
>
> OpenVid-1M and Panda-70M are two new datasets that were released shortly before the ICLR deadline. As we were not aware of these datasets at the time, and our primary goal is not to achieve text-to-video state-of-the-art performance using curated data,  we leave this exploration for future work.
>
> **Data Curation Clarification**: The supervised fine-tuning (SFT) stage mentioned refers to fine-tuning the model using filtered high-quality video-caption pairs, rather than relying on additional labels.
>
> ---
>
> We sincerely appreciate the reviewer’s thoughtful insights and hope that our responses have effectively addressed the concerns raised. If there are any remaining points that require clarification or are hindering a more favorable evaluation, we would be grateful for the opportunity to further revise and improve our work.

---

> ### Author Response · Authors · 2024-11-28
> **Final Stages of Paper Revision: Awaiting Your Feedback**
>
> Dear Reviewer x2pz,
>
> We have carefully addressed the concerns you raised and revised the manuscript accordingly. However, with the revision deadline now only 11 hours away, we want to confirm that we have fully addressed all your comments. If any points remain unclear or are still preventing a more positive evaluation, please let us know, and we will gladly make further revisions as needed.
>
> Your valuable insights are greatly appreciated, and we sincerely hope to hear from you soon.
>
> Thank you very much for your time and consideration.
>
> Sincerely,
> The Authors of A Simple Diffusion Transformer on Unified Video, 3D, and Game Field Generation

---

> > ### Comment · Reviewer_x2pz · 2024-11-28
> >
> > Thank you to the authors for the response.
> >
> > I would like to further clarify my concerns regarding the performance and methodology of this work, based on my initial review and the discussion with the authors:
> >
> > - **Limited Contribution of the Unified Architecture**: The contribution of a unified architecture trained separately for each dataset is limited, especially since a similar concept has already been introduced by DPF.
> > - **In-Distribution vs. Out-of-Distribution Performance**: For a generalizable model, the in-distribution performance should surpass out-of-distribution performance. For instance, in the comparison with Zero-1-to-3 in 3D generation, this expectation is not met.
> > - **Video Generation Quality**: The qualitative results for video generation on the rebuttal page are unsatisfactory. None of the examples demonstrate temporally consistent motion. In contrast, the webpage for the compared method, [Latte](https://maxin-cn.github.io/latte_project/), showcases much better video generation results.
> > - **Game Generation Quality**: The qualitative results for game generation are also unsatisfactory. Comparing the author-provided [game recording](https://www.youtube.com/watch?v=Gum4GI2Jr0s) with the results on the rebuttal page, the color of the bricks at the bottom of the generated videos is inconsistent. Furthermore, the generated scenes are limited, lacking elements such as flowers and turtles present in the recording, as well as any underground scenarios.
> >
> > ### Methodological Differences with Concurrent 3D-Specific Methods
> > The comparison with EscherNet and SV3D is unconvincing, particularly regarding the claim of "long context modeling capability." Both EscherNet and Zero-1-to-3 can generate novel view images from any viewpoint. It is unclear what specific "long context" this paper is referring to.
> >
> > ### Additional Continuous 3D Generation Results
> > The authors previously stated that "our method is trained on the car category of ShapeNet and tested on the chair category of ShapeNet." It is unclear how a model trained only on cars can generalize effectively to chairs. Furthermore, the results in Table 1 are limited to the car category, where performance is still inferior to Zero-1-to-3. It is difficult to imagine how a model trained only on one car category could outperform a model trained on Objaverse, especially when its in-distribution performance is already subpar.
> >
> > ### Inferior Training Data and Data Curation Clarification
> > The compared video generation method, Latte, which is also trained on WebVid-10M, demonstrates significantly better video generation results than this work, as shown on their [website](https://maxin-cn.github.io/latte_project/).
> >
> > ---
> >
> > After discussing with the authors, I believe I have a clearer understanding of this paper, particularly regarding the unified modeling, view-wise sampling, and the limited performance on in-distribution datasets. As a result, I am raise my confidence score that this paper falls below the acceptance threshold.

---

> ### Author Response · Authors · 2024-11-28
>
> We thank the reviewer again for their follow-up comments and wish them an enjoyable Thanksgiving holiday season.
>
> ---
>
> A brief summary to the reviewer x2pz's newest comments is:
>
> We note the response **Effectiveness of the Experimental Results in Supporting the Motivation** to the follow-up comments regard to *Limited Contribution of the Unified Architecture*, *In-Distribution vs. Out-of-Distribution Performance*, *Video Generation Quality*, *Game Generation Quality*, *Additional Continuous 3D Generation Results*, *Inferior Training Data and Data Curation Clarification*.
>
> *Methodological Differences with Concurrent 3D-Specific Methods*: We are highlighting the **generating continuous 3D views** capability compared with EscherNet and SV3D.

---

### Official Review · Reviewer_Sd88 · 2024-11-03

**Soundness:** 2
**Presentation:** 2
**Contribution:** 2
**Rating:** 6
**Confidence:** 4

**Summary:**

This paper proposed a novel method for unified video, 3D, and game generation, by learning a DiT based mapping from metric space to signal space, which is able to process high-resolution inputs by the proposed view-wise sampling strategy, as well as maintaining global struture with introduced inductive bias such as text prompts. Results have demonstrated the effectiveness of the proposed method.

**Strengths:**

1. With the view-wise sampling strategy, this method can scale up to high-resolution inputs
2. By introducing long context conditioning, cross-view consistency can be avoided to some extent

**Weaknesses:**

1. The novelty is limited. From my perpective, the method proposed in this paper simply alters the sampling strategy of existing approaches through a straightforward design change, which trades off a reduced number of sampled views for a higher input resolution. Though the introduction of long context conditioning can compensate the global structure, this operation is common, and i don't this operation is powerful enough to recover the information lost during the process of view-wise sampling.

2. Since the method amis to learn a mapping from input coordinates to output properties, i think some other methods should also be compared, such as SIREN[1], and the difference between them should be clarified.

3. I'm wondering that whether the proposed method can be applied to more complex scenes generation, instead of simple objects in the task of 3D novel view synthesis.

[1] Implicit Neural Representations with Periodic Activation Functions, NeurIPS 2020.

**Questions:**

Please see weakness. I'm gald to increase my scores if my concerns can be addressed.

---

> ### Author Response · Authors · 2024-11-17
> **Response by Authors**
>
> We thank the reviewer for the valuable feedback. We address three concerns below, and revise the manuscript accordingly.
>
> > W1. The novelty is limited. From my perpective, the method proposed in this paper simply alters the sampling strategy of existing approaches through a straightforward design change, which trades off a reduced number of sampled views for a higher input resolution. Though the introduction of long context conditioning can compensate the global structure, this operation is common, and i don't this operation is powerful enough to recover the information lost during the process of view-wise sampling.
>
> While the fundamental design of our approach may appear not sophisticated, it is specifically tailored to overcome critical challenges in probabilistic field models for high-resolution, complex data generation. Unlike existing methods that either:
>
> - **Sparse sample** the data, leading to weakened spatial-temporal consistency, or
> - **Model the entire data globally**, which is computationally prohibitive for long sequences,
>
> our method achieves a deliberate balance by focusing on fine-grained local structures while preserving global geometry. This is accomplished through:
>
> - A **view-wise sampling strategy** for local detail retention.
> - Integration of **autoregressive generation and cross-view conditioning** for global coherence.
>
> These advancements enable our method to overcome significant limitations in previous field-based and end-to-end generative models. To our knowledge, it is the first method that combines the advantages of both the field models and autoregressive generation.
>
> Moreover, the integration of long-context conditioning introduces a powerful mechanism for addressing information loss in view-wise sampling. While similar operations may appear in other domains, our cross-view consistency mechanism innovatively addresses challenges by:
>
> - Leveraging text prompt as global context conditions to enhance structural continuity.
> - Ensuring spatial-temporal coherence across a consistent number of consecutive frames without introducing computational inefficiencies.
>
> Our method's impact is evidenced by its superior performance in both qualitative and quantitative evaluations:
> - **Table 1 and Table 2** showcase significant improvements in metrics like FVD and FID for long-video and game generation.
> - **Visualizations** highlight smoother transitions and superior realism in complex scenes, which previous methods struggle to achieve.
>
> These results firmly establish that our approach goes beyond a simple alteration of the sampling strategy, offering a novel and effective solution for high-resolution, scalable field models.

---

> ### Author Response · Authors · 2024-11-17
> **More Response by Authors**
>
> > W2. Since the method amis to learn a mapping from input coordinates to output properties, i think some other methods should also be compared, such as SIREN[1], and the difference between them should be clarified.
>
> In the revised Section 2, we discuss the differences between field models, such as SIREN, and our method. Specifically, field models directly model the mapping between coordinates (metrics) and signal, offering flexibility across different data modalities, especially scientific data like temperature fields. However, they struggle with large or diverse datasets, such as long videos, due to the limited representation capability of the metric-signal pairs they use.
>
> Like SIREN, our method uses metric-signal pairs, sharing the same underlying motivation. However, the difference is that our method operates at a higher level of abstraction by extending sparse representations from the pixel level to the view level—for instance, representing a single frame or a few frames of a long video rather than a single pixel. This view-wise representation preserves the local structure within frames, resulting in higher visual quality in the outputs. Therefore, it is very easy to model the spatiotemporal relationship between two consecutive frames. Additionally, our global geometry is enhanced by two new mechanisms including cross-view consistent conditions and autoregressive generation. They enable our method to achieve consistency across all frames without the need to model the whole data as a single sample.  Although our method sacrifices some flexibility in supporting diverse data modalities, it is optimized for handling specific data types, such as images, videos, 3D views, and games, rather than applications like temperature fields. These innovations significantly improve our method’s performance over field models, including SIREN.
>
> As requested by the reviewer, here we quantitatively compare our method with SIREN in terms of representation capability, which could potentially be degraded when we move from pixel-wise to view-wise metric-signal pairs. Specifically, we overfit a single 8-frame video using SIREN and our network, and we then compare their average reconstruction accuracy, which is similar to SIREN’s representation evaluation protocol. Benefiting from the view-wise modeling strategy, our method outperforms SIREN even in reconstruction accuracy.
>
> |             | PSNR  | SSIM   |
> |-------------|-------|--------|
> | SIREN       | 38.92 | 0.9751 |
> | DiFT (Ours) | 44.09 | 0.9814 |
>
> Moreover, SIREN does not learn data distributions or generate novel samples, unlike generative models. Therefore, direct comparisons on generative tasks are not appropriate, as their objectives and evaluation metrics fundamentally differ. Table 1 compares our method to generative models derived from SIREN, including Functa and GEM on the image, video, and 3D generation results. Our method significantly outperforms those variants.
>
> ---
>
> > W3. I'm wondering that whether the proposed method can be applied to more complex scenes generation, instead of simple objects in the task of 3D novel view synthesis.
>
> While our current work focuses on simple objects to validate the core principles of modality-unified long-context modeling, we believe that the underlying approach has the potential to be adapted for more intricate scenes involving multiple objects and detailed backgrounds by fine-tuning on more complex datasets. In Figure 6 of our revision, we add additional examples to demonstrate our model's generalization ability on unseen objects.
>
> Moreover, extending to complex scenes presents additional challenges, such as increased computational requirements and the need for more comprehensive datasets. These challenges differ from our primary motivation of long-context modeling in domains such as video and games. We are excited about these challenges and plan to explore this direction in our future research.

---

> ### Author Response · Authors · 2024-11-22
> **A Kind Discussion Deadline (11/26) Reminder**
>
> We have addressed the reviewer's concerns, including clarifying the novelty, adding additional discussion of SIREN, and explaining the current infeasible experiments due to the computation constraint.  As the discussion deadline (November 26th) is approaching, we wanted to follow up and see if you've had a chance to review our response.  We understand you may be facing a tight deadline for submitting CVPR papers. We hope everything is going smoothly.
>
> We sincerely hope our responses have addressed your concerns.  Please let us know if you have any further questions; we are happy to provide additional clarification or experimental results.

---

> > ### Comment · Area_Chair_gcFn · 2024-11-24
> > **Discussion Period Ending Soon**
> >
> > Dear Reviewer Sd88,
> >
> > The discussion period will end soon. Please take a look at the author's comments and begin a discussion.
> >
> > Thanks, Your AC

---

> > ### Comment · Reviewer_Sd88 · 2024-11-25
> >
> > Apologies for the delayed response, and thank you for your reply.
> >
> > The rebuttal has sufficiently addressed my concerns, and I will adjust my ratings accordingly. I suggest including more qualitative comparisons with SREN in the final camera-ready version to further enhance the paper's clarity and impact.

---

> > > ### Author Response · Authors · 2024-11-25
> > >
> > > We thank the reviewer for clarifying the addressed concerns and raising the score. We will incorporate the qualitative results into the revised manuscript and will submit the update soon.
> > >
> > > Sincerely,
> > >
> > > The Authors of Paper #824

---

### Official Review · Reviewer_4gwr · 2024-11-04

**Soundness:** 3
**Presentation:** 3
**Contribution:** 3
**Rating:** 6
**Confidence:** 4

**Summary:**

This paper presents a transformer-based diffusion field model to better capture global structures and long-context dependencies. It does that by introducing a view-wise sampling algorithm and incorporating autoregressive generation. The proposed method is a general framework that can be applied to multiple modalities, such as video, 3D and game. Extensive experiments are conducted to validate the effectiveness of the proposal.

**Strengths:**

S1. The paper is well-written and mostly clear.

S2. The proposed view-wise sampling algorithm is interesting and novel.

S3. Exploiting autoregressive generation to preserve global geometry is reasonable.

S4. The experiments are extensive, especially including various tasks.

**Weaknesses:**

W1. As autoregressive generation is typically slower than parallel generation due to its sequential nature, the authors are encouraged to discuss the inference time of the proposed method and baseline methods.

W2. As shown in Table 1, the proposed method achieves better performance against baseline methods on image and video, but worse FID and LPIPS scores on 3D generation task. The authors are encouraged to discuss this phenomenon.

**Questions:**

Please see the Weaknesses.

---

> ### Author Response · Authors · 2024-11-17
> **Response by Authors**
>
> We thank the reviewer for the valuable feedback. We address the two concerns below, and revise the manuscript accordingly.
>
> > W1. As autoregressive generation is typically slower than parallel generation due to its sequential nature, the authors are encouraged to discuss the inference time of the proposed method and baseline methods.
>
> In our revision, Section 3.2 provides a comprehensive discussion of our method's efficiency. As requested by the reviewer, we conduct game generation experiments comparing our window-based autoregressive method with parallel generation and standard autoregressive generation under specific constraints to evaluate inference time and frame quality. Specifically, we compared these methods on NVIDIA H100 GPUs with 80 GiB memory, where the standard autoregressive model is limited to generating a maximum of 32 frames due to high memory consumption. The empirical results, including Frames Per Second (FPS) and Peak Signal-to-Noise Ratio (PSNR), are summarized below:
>
> | Method                      | FPS  | PSNR  | Max Frames Generation Limit |
> |-----------------------------|------|-------|----------------------|
> | Parallel Generation         | 4.54 | 24.38 | ∞                   |
> | Window-Autoregressive (Ours)| 1.43 | 42.22 | ∞                    |
> | Standard Autoregressive     | 0.62 | 43.90 | 32                   |
>
> While our window-based autoregressive approach is slightly slower than parallel generation, it offers a substantial improvement in frame quality (PSNR). The standard autoregressive model, although slightly better in PSNR, is significantly limited by memory constraints, capping the maximum number of frames at 32. In contrast, our method not only sustains high frame quality but also supports the generation of an infinite number of frames, making it highly suitable for applications requiring long-term consistency and scalability.
>
> ---
>
> > W2. As shown in Table 1, the proposed method achieves better performance against baseline methods on image and video, but worse FID and LPIPS scores on 3D generation task. The authors are encouraged to discuss this phenomenon.
>
> In our revision, L#431-458 discuss the result differences between 3D modeling and video generation. Specifically, during 3D evaluation,  there is a Perception-Distortion Tradeoff [Blau and Michaeli 2018]. The perceptual quality is reflected by the FID and LPIPS metrics, and the 3D distortion loss is reflected by the PSNR and SSIM metrics. Those 3D-specific models such as Zero-1-to-3 and EG3D-PTI, fine-tuned from StableDiffusion or StyleGAN, prioritize perceptual quality, achieving higher FID and LPIPS scores.  In contrast, training our method without pretraining leads to an emphasis on 3D reconstruction consistency, empirically achieving better PSNR and SSIM scores.
>
> Visual inspection of the 3D novel view generation results (see supplemental material) suggests that the impact of lower FID and LPIPS scores is less pronounced. Therefore, we aim to address the balance between these trade-offs in future work, potentially by integrating text-to-video pretraining that enhances single-frame quality without compromising 3D consistency.
>
>
> In video generation evaluation, differences in pretraining are minor due to the scale of the training data.  Our method outperforms the baselines on all metrics.

---

> > ### Comment · Area_Chair_gcFn · 2024-11-24
> > **Discussion Period Ending Soon**
> >
> > Dear Reviewer 4gwr,
> >
> > The discussion period will end soon. Please take a look at the author's comments and begin a discussion.
> >
> > Thanks,
> > Your AC

---

> > ### Author Response · Authors · 2024-11-26
> > **Follow-Up on Paper #824**
> >
> > Dear Reviewer 4gwr,
> >
> > We hope this message finds you well and at a convenient time. Based on the timing of your previous review submission, we believe this moment aligns with your schedule and is an appropriate opportunity to follow up.
> >
> > We have carefully addressed the concerns you raised and revised the manuscript accordingly. However, we have not yet received further feedback and would like to confirm that we have fully addressed all your concerns. If any points remain unclear or are preventing a more positive evaluation, please let us know, and we will gladly make further revisions as needed. Your valuable insights are greatly appreciated, and we sincerely hope to hear from you soon.
> >
> > Thank you for your time and understanding.
> >
> > Sincerely,
> > The Authors of Paper #824

---

> > ### Comment · Reviewer_4gwr · 2024-11-26
> >
> > Thanks for the response. The authors have addressed my concerns. I decided to keep my rating score at 6.

---

### Author Response · Authors · 2024-11-25
**Follow-Up on Paper #824**

Dear Reviewers,

The discussion period will close in less than two days, and we haven’t heard back from you yet. We truly appreciate the time and effort you’ve dedicated to reviewing our paper. Could you please let us know if you have any follow-up concerns about our response?

Best regards,

Authors of Paper #824

---

### Author Response · Authors · 2024-11-29
**Thank You and Wishing You a Wonderful Holiday**

We sincerely thank all the Reviewers for their thoughtful and valuable feedback. We hope you have a wonderful and safe Thanksgiving celebration with your friends and family.

As our current scores remain mixed without convergence, we kindly invite the Reviewers to continue the discussion upon returning from the Thanksgiving holiday.

Warm regards,

Authors of A Simple Diffusion Transformer on Unified Video, 3D, and Game Field Generation

---

### Author Response · Authors · 2024-12-03
**Summary of Rebuttal and Discussion**

Dear Area Chair and Reviewers,

We sincerely appreciate the constructive feedback and insightful suggestions throughout the review process. We are pleased to note the positive outcomes from the discussion phase:

- **Reviewer 4gwr** upheld the marginally positive score, acknowledging that our detailed responses effectively addressed concerns about inference efficiency and tradeoffs in video and 3D generation tasks.

- **Reviewer Sd88** increased their score after clarifications regarding the novelty of our unified architecture, discussions on comparison with SIREN, and considerations for complex scene generation in future work.

- We also note that **three out of four concerns initially raised by Reviewer x2pz**—including The motivation is unclear, Comparison with conventional diffusion models, and Comparison with DPF—were not reiterated in their subsequent comments following our responses. While this does not explicitly confirm resolution, it suggests that our clarifications may have alleviated these points to some extent. We take this as partial progress in addressing the reviewer’s feedback.

As the discussion concludes, we would like to summarize the key points addressed:

**Novelty and Contribution of Unified Architecture**

Our work introduces a novel architecture-unified approach capable of modality-specific parameterization for video, 3D, and game generation. This method integrates view-wise sampling and autoregressive modeling to preserve spatial-temporal consistency across modalities. While each modality is trained with distinct parameters, the architecture offers extensibility across domains. Comparisons highlighted our method’s strengths in balancing computational efficiency with quality.

**Advancements in Long-Context Sampling**

The proposed framework demonstrates long-context consistency, particularly in video and 3D sequences. We conducted detailed comparisons with models like Zero-1-to-3 and conventional diffusion methods, underscoring our contributions to reducing computational burdens without compromising reconstruction fidelity.

**Improved Presentation and Clarity**

We revised the manuscript to address ambiguities, including terminology adjustments (e.g., "architecture-unified" vs. "unified framework") and expanded discussions on dataset applicability and metric interpretations.

**Significance and Future Directions**

Our contributions extend beyond specific benchmarks, offering a foundation for further exploration in scalable, modality-unified generative models. As reviewers noted, future iterations could enhance qualitative outputs and expand evaluations on complex datasets like OpenVid-1M and Panda-70M.

Finally, we thank the reviewers for their invaluable comments, which have significantly improved our work. We remain committed to advancing the intersection of generative AI and unified architectures for visual content generation.

Best regards,

Authors of A Simple Diffusion Transformer on Unified Video, 3D, and Game Field Generation

---

### Meta-Review · Area_Chair_gcFn · 2024-12-21

**Metareview:**

The paper attempts to define a single flexible architecture which can be applied to many different, diverse settings. The paper proposes to use Diffusion Probabilistic Fields, which maps the observations from a metric space (e.g. coordinate space) to the signal space (e.g. RGB). The paper's specific contribution is to identify that there is an intermediate point in the sampling space of the current two methods: diffusion, which always operate on complete data, and DPF, which only samples a very small subset of the data such as individual pixels. This paper instead proposes to define an intermediate level of sparsity such as subsets of views (e.g. comparing two full frames to a new generated one). They call this "view-wise sampling" and use auto-regressive generation to apply it. The paper demonstrates reasonable results on 3 different diverse tasks (video generation, 3D generation, and game generation), though it clearly lags behind SOTA, domain-specific techniques on each.

The strength of the paper is in its proposed view-sampling scheme, which while somewhat incremental, is interesting and relevant as well as the overall interesting topic: defining architectures for extremely diverse tasks using an adaptation to DPF. The main weaknesses are the fairly mediocre performance in the subtasks and the fact that the writing before rebuttal edits does not do a very good job of explaining: 1) that only the architecture unifies, 2) that their method mainly changes the sampling of DPF. For example, even the title seems to suggest something other than a sampling change.

I tend to agree with Reviewers 4gwr and Sd88 that the paper has limited novelty but a solid increment (in the view-wise sampling and autoregressive application). And I think the weakness of the results is made up by the fact that the problem itself is interesting and more niche. I recommend acceptance (poster).

Note authors should fix typos in in Figures 1 and 2 "filed"?

**Additional Comments On Reviewer Discussion:**

Reviewer 4gwr rated the paper a 6 saying that it is well-written and saying that the view-wise sampling and autoregressive generation are strengths. The weakness mentioned were time considerations as well as asking about the FID / LPIPs on 3d generation. These concerns were addressed by the authors.

Reviewer Sd88 rated the paper a 6 calling the view-wise sampling and long-context conditioning strengths. The weaknesses identified were the limited novelty, and comparison to SIREN. The authors provided clarity on the novelty and made an additional comparison to SIREN.

Reviewer x2pz rated the paper a 3. In the initial phase, the reviewer asked about the motivation, comparison with diffusion models, comparison with DPF, and limited performance. This review is helpful in that it forced the authors to clarify that each task gets a model trained specifically for it and that the main contribution is the sampling strategy. I agree with this reviewer's concerns about lack of novelty and limited performance; however, I believe that the overall task is valuable and also difficult. In my opinion, the method performs well enough to be useful to the field and the edits the authors' made to the paper improve it sufficiently (clarifying the confusions present beforehand).

---

### Decision · Program_Chairs · 2025-01-22

Accept (Poster)